# RealCity3D: A Large-scale Georeferenced 3D Shape Dataset of Real-world Cities

**Congcong Wen** [1*]    **Wenyu Han** [1*]    **Lazarus Chok** [1†]    **Yan Liang Tan** [1†]
**Sheung Lung Chan** [1†]    **Hang Zhao** [2]    **Chen Feng** [1‡]
[1]New York University    [2]Tsinghua University
https://github.com/ai4ce/RealCity3D

## Abstract

Existing 3D shape datasets in the research community are generally limited to objects or scenes at the home level. City-level shape datasets are rare due to the difficulty in data collection and processing. However, such datasets uniquely present a new type of 3D data with a high variance in geometric complexity and spatial layout styles, such as residential/historical/commercial buildings and skyscrapers. This work focuses on collecting such data, and proposes city generation as new tasks for data-driven content generation. Thus, we collect over 1,000,000 geo-referenced 3D building models from New York City and Zurich. We benchmark various baseline performances on two challenging tasks: (1) city layout generation, and (2) building shape generation. Moreover, we propose an auto-encoding tree neural network for 2D building footprint and 3D building cuboid generation. The dataset, tools, and algorithms will be released to the community.

## 1 Introduction

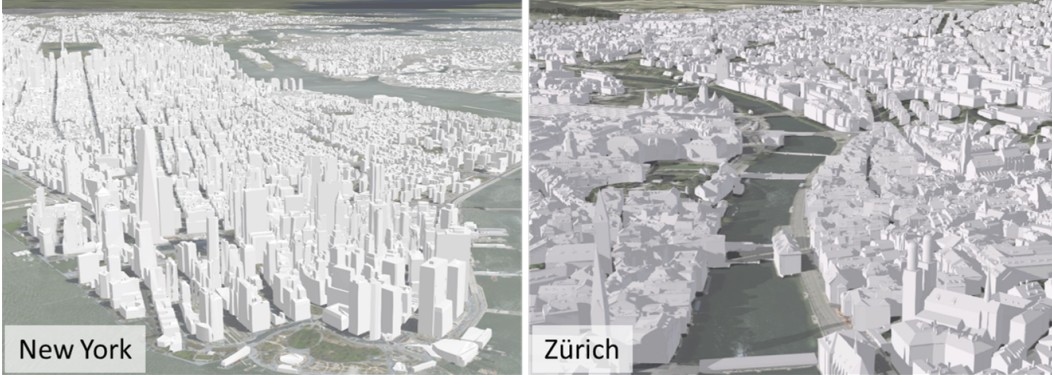

Figure 1: Overview of New York and Zurich cities in our RealCity3D dataset.

As an important arena for human activities, cities have been a focal point of research. Alongside the rapid advancement of image/video generation, data-driven 3D city generation has become more feasible and appealing because of 1) the increasing availability of city-level remote sensing, and

---

[*]Equal contributions.

[†]Equal contributions.

[‡]The corresponding author is Chen Feng cfeng@nyu.edu.

Submitted to the 35th Conference on Neural Information Processing Systems (NeurIPS 2021) Track on Datasets and Benchmarks. Do not distribute.

2) the intensification of data-driven methods in architecture and urban planning. Urban planners increasingly rely on city-level simulations to make planning decisions; game designers use city generation tools to automatically generate virtual city environments; and more recently, there is a surging demand from the autonomous driving industry to conduct road testing in simulated 3D environments. All of these potential applications have increased demand for realistic city generation.

While deep generative models are successful for various data modalities, including language, audio, image, video, and even point clouds, several difficulties prevent deep generation from being applied towards city-level geometric generation. First, cities are a set of complex geometrically parameterized objects with irregular layouts. Second, these objects usually live on a high-dimensional complex data manifold. For example, a building object records as a set of 3D polygons, each contains a variable number of 3D vertices. More importantly, there are few publicly available 3D real-world city datasets, which are essential for developing data-driven methods, particularly deep generative models.

Existing public datasets for geometric data generation can be categorized at the object-level [5], home-level [1, 27], and city-level [7, 2]. However, it is difficult to achieve city-level generation by training models on object/room-level datasets due to their limited scale. Moreover, it is unrealistic for trained models to generate 3D cities on existing city-level datasets due to the restriction of data dimension, as they mostly contain 2D data such as polylines and polygons. To overcome these challenges, some researchers [6] have developed their own synthetic datasets since no public datasets are available from real-world city buildings. To a large extent, the lack of viable, real-world 3D city datasets have hindered the development of deep generative models for city-level generation.

To this end, we propose the RealCity3D dataset, a real-world, city-level 3D dataset for New York City and Zurich (Figure 1). The dataset consists of over 1,000,000 georeferenced objects covering a total area of more than 871.7 square kilometers. Four different representations are provided per object: polygon mesh, triangle mesh, point cloud, and voxel grid. Semantic information of objects are preserved in polygon meshes. Based on RealCity3D, we explore the possibility of city generation. Considering the difficulty of the problem and the absence of applicable methods, we split the city generation task into two subtasks: *Task 1 City layout generation* and *Task 2 Building shape generation*.

Our contributions are three-fold: (1) We open-source a large-scale georeferenced 3D shape dataset RealCity3D, in multiple forms including polygon meshes, triangular meshes, point clouds and voxel grids; (2) We perform city/building scale generation benchmarks to explore the research directions RealCity3D can support; (3) We propose a simple but efficient tree neural network that encodes and generates spatial data hierarchically for 2D building footprint and 3D building cuboid generation.

## 2  Related work

### 2.1  Datasets for Geometric Generation

Current art in city-level geometric generation are mostly based on synthetic datasets that do not have the same geometric complexity and diversity that real-world cities have. Publicly available, city-level object datasets that are georeferened to real world cities are rare; they can be divided into three categories according to its scale: **1) Object-level Datasets**: ShapeNet [5] contains over three million 3D models with a core dataset of about 51,300 unique 3D models across 55 common object categories. Though some studies [10, 3, 16] achieve promising single object generation performance, it will be difficult to extend trained models on this synthetic dataset to a large-scale, real-world dataset. On the contrary, RealCity3D deals exclusively with city-scale building objects georeferenced to the real world. **2) Room-level Datasets**: LIFULL HOME's database [1] contains five million floor plans. RPLAN [27] consists of 80,000 floor plans from real-world residential buildings. HouseGAN [17] and Graph2Plan [13] perform indoor layout generation on these two datasets respectively. However,

Table 1: Comparison with representative datasets

| | Datasets | Year | Spatial extent | Objects | Format | Generation Task |
|---|---|---|---|---|---|---|
| Object-level | ShapeNet [5] | 2015 | - | 3,000,000+ | Mesh | 3D Object Generation |
| Room-level | LIFULL HOME [1] | 2015 | - | 5,300,000+ | Imagery | 2D Indoor Layout Generation |
| | RPLAN [27] | 2019 | - | 80,000+ | Imagery | 2D Indoor Layout Generation |
| City-level | RoadNet [7] | 2019 | $1.7 \times 10^8 m^2$ | - | Imagery/Polylines | 2D Road Network Generation |
| | SpaceNet v2 [2] | 2018 | $30.11 \times 10^8 m^2$ | 685,000 | Imagery/Polygons | 2D City Layout Generation |
| | **RealCity3D** | 2021 | $8.71 \times 10^8 m^2$ | 1,000,000+ | Mesh/Point Cloud/Voxels | 3D Object and 2D/3D City Generation |

the number of rooms in each floor plan rarely exceeds thirteen, limiting the extensive ability of trained models from being applied to city-scale generation tasks. **3) City-level Datasets**: RoadNet [7] is a real-world road network dataset collected from OpenStreetMap (OSM) of 17 cities; SpaceNet [9] offers over 685,000 building footprints across 5 cities. These two datasets only involve 2D polyline or polygon data, which are limited to a certain extent considering the complexity of 3D real-world applications. By contrast, RealCity3D is a city-level dataset that consists of not only 2D city layout information, but also building facade details. This enables more complex, large-scale city generation tasks. We compare the statistics of RealCity3D with some existing datasets in Table 1.

## 2.2 City Generation Methods

Existing work on city generation focuses on non-data-driven methods, i.e., procedural modeling. Due to the lack of high-quality 3D city training data, few techniques have been proposed to achieve data-driven 3D city generation. RealCity3D changes this: we demonstrate in our benchmarks how deep generative neural networks can benefit from our enriched, multi-format 3D city-level datasets.

**Procedural modeling**, such as L-systems, create geometric structures based on handcrafted shape grammar [15, 26, 31, 8], a set of Euclidean shape transformation rules. However, handcrafted modeling becomes extremely intensive when urban designs become more complex and diverse. To automatically learn these rules, inverse procedural modeling uses deep neural nets to extract shape grammar from existing 2/3D datasets [25, 22, 18, 11]. The ESRI CityEngine, popular with the urban planning community, is a commercially available generation engine to create and apply shape grammars to generate large-scale city layouts. However, these procedural generation engines require experts to manually adjust the rules and parameters. Data-driven methods which can automatically learn the features and rules of city generation, with limited human input, are more appealing.

**Data-driven generation methods** have gained popularity in recent years as it enables the generation of complex geometric structures (vertices/lines/surfaces) with minimal human input. 1) For single object generation, [16] proposed PolyGen to generate 3D polygon meshes with an autoregressive transformer model. PolyGen's code is not open-sourced and cannot be benchmarked. Instead, we perform building shape generation using Raw-GAN and Latent-GAN [3] on our RealCity3D point cloud dataset. 2) For layout generation, House-GAN by [17] is a GAN-based indoor layout generator. However, this graph-based method cannot be directly performed on our dataset. Other city-level layout generators like [7]'s Neural Turtle Graphics (NTG) for road network generation, and [32]'s image-based supervised architecture reconstruction are incompatible with RealCity3D's object-type dataset. For generating sequential sketch strokes, [12] proposed SketchRNN, an RNN model with a VAE structure. By converting each city layout to a sequential list of points (like a sketch), SketchRNN could be used as a city layout generation benchmark on RealCity3D.

## 3 Task Definition and Data Processing

### 3.1 Task Definition

RealCity3D is a large-scale georeferened 3D shape dataset of real-world city buildings represented in four common formats, i.e. polygon mesh, triangular mesh, point clouds, and voxels. This is the first-of-its-kind dataset that will enable the research community to develop new data-driven techniques for large-scale city generation tasks. The ideal dataset for city generation should be georeferenced, have comprehensive coverage of whole cities, be available in common geometric representations (e.g. meshes, point clouds, voxels) to allow for a variety of training approaches, and contain geometric complexity (or architectural LoD - Level of Detail, see Figure 2). Such a dataset will enable 3D city generation, containing both realistic city layouts and building shapes/facades. However, considering the difficulty of problem and that no existing methods have been developed for 3D city generation, we simplify the city layout generation at LoD2 complexity to LoD1 complexity by transforming each building polygon into a footprint in 2D space and minimum bounding cuboid in 3D space. We treat LoD1 city layout and LoD2 building shape generations as separate tasks; the former focusing on the spatial distribution of 2/3D polygons, and the latter focusing on 3D object geometry:

**Task 1: City layout generation**. City layouts refer to the spatial distribution and shapes of 2D building footprints or 3D buildings in urban space. Generating new city layouts with reference to existing urban forms is a complex but valuable task within the urban planning and game design

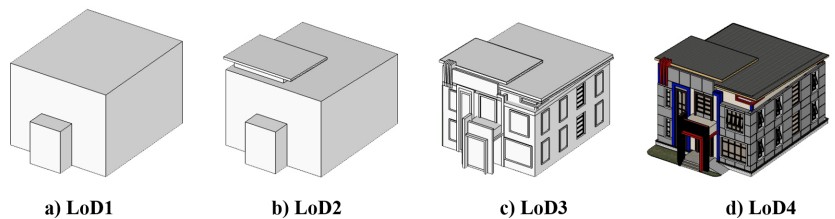

a) LoD1      b) LoD2      c) LoD3      d) LoD4

Figure 2: Examples of the 3D building object ranging from LoD1 to LoD4. LoD4 sketch retrieved from SketchUp 3D Warehouse.

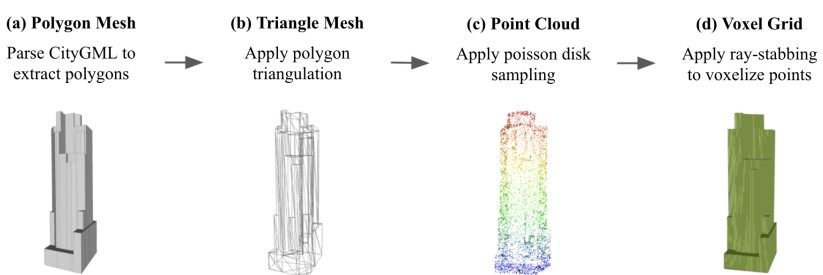

Figure 3: Overview of RealCity3D dataset creation and data transformation on a single LoD2 building. (a) Polygon mesh and (b) triangle mesh visualised in MeshLab, (c) point cloud in CloudCompare with colorized z-values, (d) voxel grid in viewvox.

communities. In urban planning, city layouts affect urban microclimates, land-use patterns and urban transportation networks. RealCity3D allows for the development of new deep learning approaches that can generate large-scale city layouts with greater LoD complexity, efficiently and accurately.

**Task 2: Building shape generation**. Building shapes refer to the external 3D facade of each building polygon. Modelling the 3D building envelope with high architectural detailing (i.e. LoD2 and above) is a challenging task due to its geometric complexity, but essential for generating realistic 3D cities.

## 3.2 Data Standard and Collection

In our dataset, 3D building objects data are extracted from 3D city models in CityGML format, a XML-based format widely used by the AEC community for efficient storage of city-scale data. CityGML extends XML by adding sets of primitives, including topology, features, and geometry, as well as city-specific constraints. Examples of 3D object classes in CityGML include buildings, tunnels, and bridges. CityGML has a hierarchical model complexity system to mark the complexity of each object class from LoD1 (Levels of Detail) to LoD4, as shown in Figure 2.

We collected CityGML data of New York City and Zurich from The New York City Department of Information Technology [24] and Stadt Zurich [23] respectively. Since CityGML data from publicly available 3D geospatial datasets contain building models mostly with LoD2 complexity, we store all building objects in our dataset in LoD2 format. Besides, the CityGML data quality of different cities in our datasets vary considerably, presenting technical difficulties for scalable data processing. For example, only 76% of Zurich buildings have valid CityGML surfaces, and the other 24% have non-planar duplicated surfaces which violate the CityGML format standard. Our data processing pipeline, described below, can be scaled across cities where CityGML data are available.

## 3.3 Dataset Processing

### 3.3.1 Parsing CityGML Files

From each CityGML file, we extract building polygons and their semantic information as dictionaries. Only exterior components (i.e. polygon surfaces) are conserved. Recurring polygon vertices are removed, ensuring triangulation can be performed without error. Each building surface is categorized

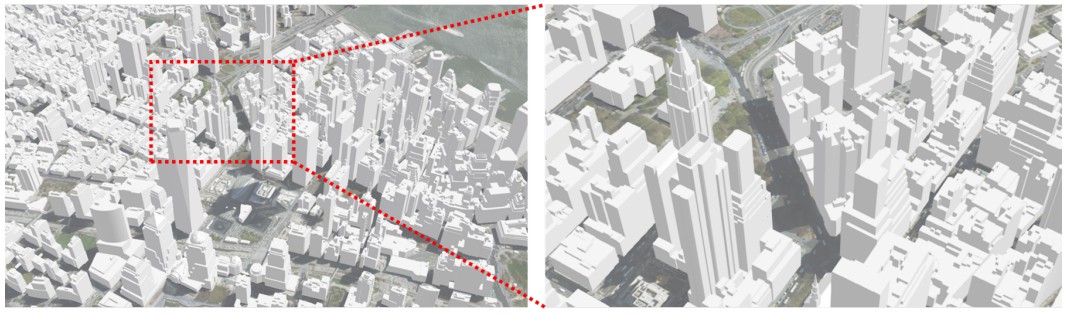

Figure 4: Detailed views of 3D building shapes in New York

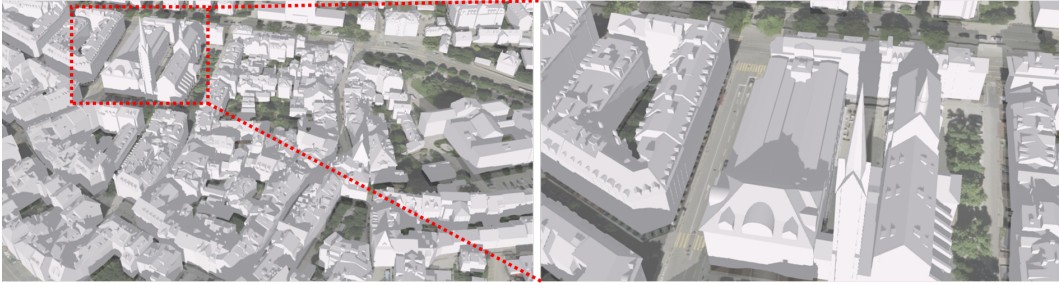

Figure 5: Detailed views of 3D building shapes in Zurich

as "GroundSurface", "RoofSurface", and "WallSurface" based on its CityGML building semantic information. Each semantically labelled polygon is output as an .obj file for further transformations.

### 3.3.2 Polygon Triangulation

Taking each polygon mesh building data, we apply polygon triangulation to decompose each polygon area **P** into a maximal set of non-intersecting triangles on a continuous surface. The union of these non-intersecting triangles is **P**, with each diagonal line segment connecting two vertices of **P**. The triangulation of each polygon with *n* vertices consists of exactly *n-2* triangles.

### 3.3.3 Point Sampling

We acquire 3D point clouds from each triangular mesh using Poisson disk sampling, a sequential, bias-free process for selecting points in each triangular subdomain. Poisson disk sampling has been used to achieve approximately uniform distance between adjacent points, yielding good visual resolution for rendering 3D buildings [29]. By uniformly sampling these points on a continuous mesh surface, we reduce the amount of noise/outliers that may come with conventional LiDAR scans of city buildings. Our uniformly dense point clouds suit deep learning approaches such as voxel-based convolution neural networks and deep learning on unstructured point clouds (e.g. PointNet) [4].

### 3.3.4 Voxelization

To provide greater geometric structure, we organise the 3D point cloud into a discrete voxel representation. Voxelization is a common method for downsampling and facilitating rapid retrieval of large-scale point cloud data [28], as would be essential in real-world city planning applications. We use the open source binvox program to efficiently rasterize the point cloud into a 3D voxel grid, which uses a variation of the ray-stabbing method described in [19]. The ray-stabbing method classifies voxels as either as an interior or exterior voxel by imagining a ray stabbing through the mesh model. Voxels at the two extreme depth samples of the ray (i.e. when the ray first penetrates the model, and when it leaves the model) are classified as exteriors; otherwise, they are classified as interiors.

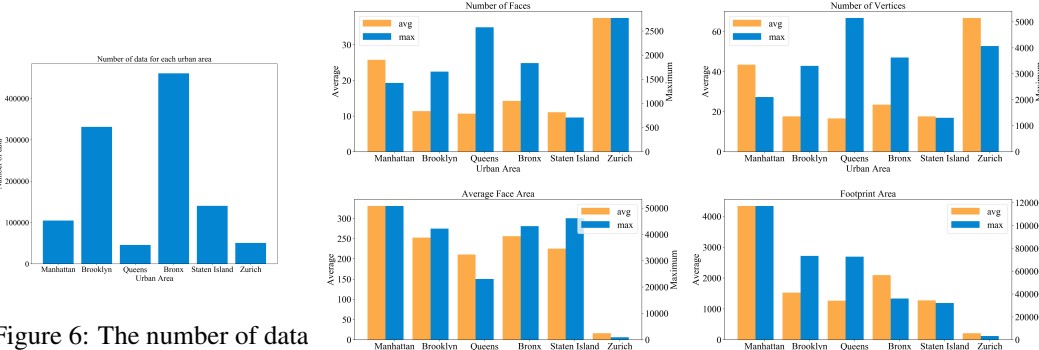

Figure 6: The number of data for each urban area

Figure 7: Building Mesh Statistics.

## 3.4 Data Statistics

New York City is an amalgamation of five different boroughs which have their own unique architecture due to its rich architectural history and land-use patterns. We divided the NYC dataset into its five boroughs: Manhattan, Brooklyn, Queens, Bronx and Staten Island. In total, we extracted $1,133,813$ individual building models with polygon meshes, triangular meshes, point clouds and voxels representations. The number of models in different boroughs is shown in Figure 6. Building mesh statistics are shown in Figure 7 to demonstrate the geometric complexity and variance expressed in one dataset. As can be seen from the number of vertices and faces, some building shapes are highly complex with thousands of faces, while others have far fewer, adding learning challenges.

## 4 Dataset Benchmarks

### 4.1 Task 1: City Layout Generation

We first evaluate the existing methods on the 2D city layout generation. Here, we benchmark two types of methods: procedural modeling via CityEngine and several well-characterized data-driven methods. Using CityEngine, an urban planner manually tuned the rules and parameters to generate city layouts based on RealCity3D data. As we can always make the results quantitatively perfect after time-consuming, hand-tuning of parameters, we decided that it is unfair to quantitatively compare the CityEngine-derived layouts with other data-driven benchmarks. These data-driven methods can be summarized into: (1) point clouds set-based methods, such as PointNet [20] and PointNet++ [21], which treat the city layout data as a set of point clouds; (2) sequence-based methods, such as SketchRNN [12], which regard the layout data as a sequence. In addition, we propose a tree-based method for constructing the layout data as a hierarchical tree. In the experiments, 45,487 buildings in Manhattan borough are selected and batched into sets of 32 neighboring buildings. 70% of the data are taken as training sets, 10% as valiation sets and 20% as test sets. The learning rates for SketchRNN-R2 method, SketchRNN-R5 method, PointNet-MLP method, PointNet2-MLP, and AETree are $0.001, 0.0001, 0.001, 0.001, 0.001$ respectively, and the batch sizes for these methods are $100, 100, 500, 500, 50$ separately. All the baseline methods run on an NVIDIA GeForce GTX 1080 Ti GPU. The model that achieves the best 2D generation performance is selected for 3D building cuboid generation to further demonstrate the results of 3D city layout generation.

### 4.1.1 Evaluation Metrics

We use three popular metrics proposed in [3] to evaluate generation. *Jensen-Shannon Divergence* (JSD) measures the similarity of marginal distributions between reference and generated sets. The distribution of data is calculated by counting the points in each discretized grid cell. *Coverage* (COV) measures the fraction of points in generated data that are matched to the corresponding closest neighbor points in the reference data. *Minimum Matching Distance* (MMD) measures the fidelity of a generated set with respect to a reference set by matching each generated point to the point in reference data with the minimum distance. MMD is the average of distances in the matching.

For COV and MMD, we only select *Chamfer Distance* (CD) to compute the distance between two point clouds. We leave out *Earth Mover's Distance* (EMD) as it requires the number of instances in two sets to be equal, which is not suitable for our generation evaluation.

We introduce *Overlapping Area Ratio* (OAR) to measure the extent of overlapping in generated layouts. Different from *Intersection over Union* (IoU), OAR measures the ratio of objects' area that have overlapped with others to all objects' area, instead of the ratio of intersection area among objects to the union of all objects' area, which can be defined as:

$$OAR(O) = \frac{\sum_{o \in O} A(o), \quad if(o \cap \hat{o}), \forall \hat{o} \in \{O - o\}}{\sum_{o' \in O} A(o')}, \tag{1}$$

where A(·) is the object $o$ area, $\cap$ indicates two overlapping objects, $O$ is the set of generated objects.

### 4.1.2 Baselines

**CityEngine**. ESRI CityEngine is a commercial software that uses a procedural modeling approach based on L-systems to create large-scale city models. This approach is different from deep generative models that are data-driven. By creating road networks and dividing the parcels into lots, it generates buildings on the allotments using predefined rules and parameters. The building footprint is generated using default rules with some manual adjustment of parameters.

**SketchRNN-R2**. SketchRNN is a generative model to generate sketch drawings [12]. This model seems intuitively suitable to solve our problem. We convert each city layout data to a list of points as a sketch with x,y coordinates according to the input of SketchRNN. Specifically, for a batch with 32 building footprints, the converted sketch consists of 128 points by taking all corners.

**SketchRNN-R5**. Based on vanilla SketchRNN, we explore replacing the parameter of a city layout data (i.e. the center coordinates, length, width, height and orientation angle ) with x,y coordinates of a sketch. So we transform a batch of data to a sketch with 32 high dimension points, which incorporates 5 elements: ($\Delta x$, $\Delta y$, $\Delta l$, $\Delta l$, $\Delta a$). The first five elements are the offset parameters from the previous box. Different from [12], we use 1D to represent the binary state of the pen (at its end or not), since we assume that the pen draws 32 points in succession.

**PointNet-MLP**. In addition, we benchmark a simple baseline model, which adopts PointNet [20] as the encoder by regarding a city layout (corner points) as a point cloud. By reference to the decoder of SketchRNN, we employ MLP to decode the latent representations to parameters for a probability distribution of points. Meanwhile, the loss function aims to maximize the log-likelihood of the generated probability distribution to explain the training data.

**PointNet2-MLP**. Moreover, we replace the PointNet with PointNet++ [21] as model encoder while keeping the same experimental settings as PointNet-MLP.

**AETree**. Lastly, we propose AETree, a tree structured neural network that efficiently encodes and generates areal spatial data hierarchically. A tree encoder with LSTMCell learns to extract and merge spatial information from bottom-up iteratively. The resulting global representation is reversely decoded for reconstruction or generation. More details can be found in the Appendix.

### 4.1.3 Results

Table 2: Quantitative comparisons of city layout generation performance with various data-driven baseline methods. The first four columns represent the results of models under four generation evaluation metrics and the last two columns measure the complexity of models.

| Methods | JSD($\downarrow$) | COV(%, $\uparrow$) | MMD($\downarrow$) | OAR(%, $\downarrow$) | #params | FLOPs/sample |
|---|---|---|---|---|---|---|
| SketchRNN-R2 | 0.0089 | 33.62 | 0.0050 | 1.83 | 2.19M | 243.13M |
| SketchRNN-R5 | 0.0101 | 28.76 | 0.0047 | 95.41 | 2.37M | 402.46M |
| PointNet-MLP | 0.0417 | 4.60 | 0.0219 | 87.47 | **1.84M** | **3.67M** |
| PointNet2-MLP | 0.0407 | 22.36 | 0.0086 | 56.39 | 2.03M | 31.55M |
| AETree | **0.0033** | **39.53** | **0.0044** | **1.66** | 2.91M | 31.86M |

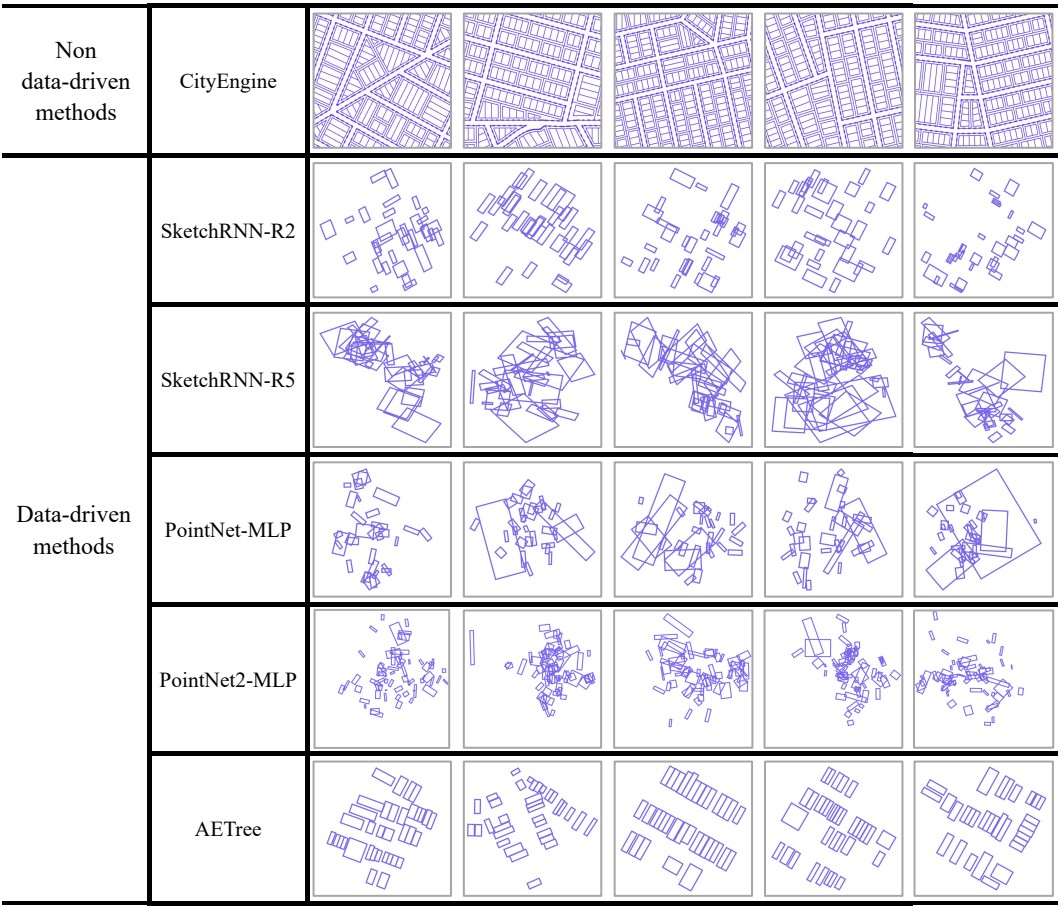

Figure 8: City layout generation results of the models trained on NYC dataset.

We quantitatively compare the city layout generation results in Table 2. Baseline models do not perform well across the four evaluation metrics, in comparison with the proposed AETree model. To intuitively show model performance, we randomly select some generation results of each model, as shown in Figure 8. Generated layouts from the SketchRNN and AEtree model are more regular than the other three data-driven methods. CityEngine is able to generate well-ordered city layouts, but loses style variance due to its rigid set of predetermined parameters and rules.

### 4.1.4 Discussion

The results demonstrate that most data-driven baseline models do not perform well on our city layout generation dataset. While the proposed AETree model generates reasonable city layout results, there is still significant room for improvement on 3D minimum bounding cuboid data (see Figure 9). Furthermore, both 2D building footprints and 3D minimum bounding cuboids are generated by simplifying the original LoD2 to LoD1. We expect that it will be difficult to achieve detailed city generation based on existing methods, thus we divided this problem into two sub-tasks. We invite the research community to develop novel city layout generation methods using RealCity3D.

### 4.2 Task 2: Building Shape Generation

One unique property of building object of RealCity3D is being geometrically highly constrained. For example, most buildings have vertical walls and planar surfaces, but some involve highly intricate facade details (e.g. the Empire State Building compared to a town house). This would be challenging for existing 3D deep learning models to learn features from building objects. To demonstrate this,

Table 3: Benchmark of point cloud generation on two datasets.

| **RealCity3D** | JSD | Coverage | MMD |
|---|---|---|---|
| Raw-GAN | 0.068 | 47.6 | 0.061 |
| Latent-GAN [3] | 0.024 | 57.3 | 0.088 |
| **ShapeNet** [5] | JSD | Coverage | MMD |
| Raw-GAN | 0.176 | 52.3 | 0.0020 |
| Latent-GAN [3] | 0.020 | 68.9 | 0.0018 |

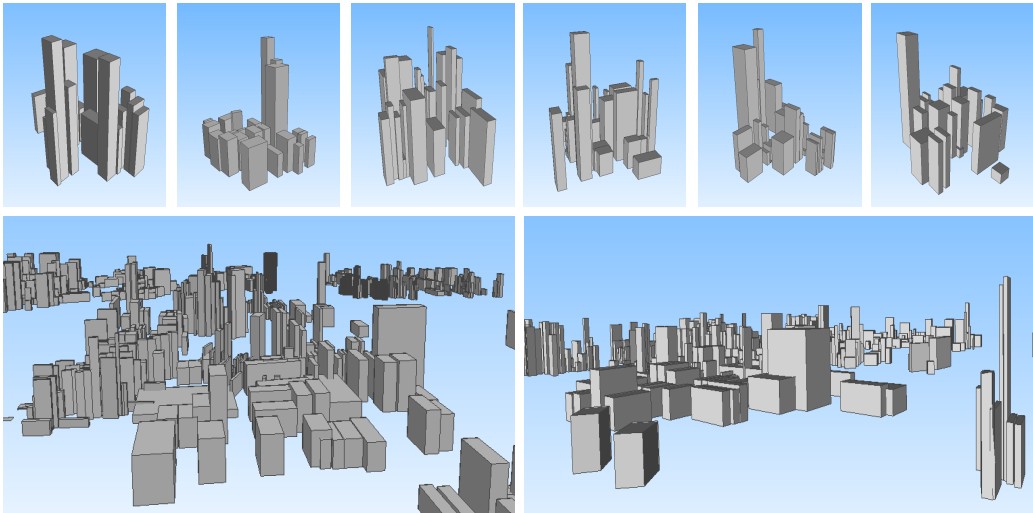

Figure 9: 3D generation results of AETree trained on the NYC Dataset

we train FoldingNet on the point cloud representations of RealCity3D, and qualitatively show the challenges it faces in reconstructing the 3D building shapes (Appendix Figure 16).

To quantitatively evaluate the difficulty of this task, we choose to train Raw-GAN and Latent-GAN for building shape generation on the point cloud representation of our datasets. We also perform the same experiments on ShapeNet [5], a simpler object-level dataset, and compare the generation results using JSD, COV, and MMD metrics (metric descriptions in Section 4.1.1). We report results in Table 3. The same generative models perform differently on the two datasets, indicating the two datasets have different properties relating to 3D shapes generation. The visualized generation results of Latent-GAN are shown in Appendix Figure 15. It can be seen that the reconstructions lost many important geometric details and variations of the 3D building shapes. Hence, the uniqueness of RealCity3D and the additional learning challenge it poses to 3D computer vision community are further demonstrated.

## 5   Conclusions

In this paper, we introduce RealCity3D, a large-scale georeferenced 3D shape dataset of real-world cities, including New York City and Zurich. The dataset covers more than 871.7 square kilometers and consists of over 1,000,000 georeferenced objects, which are represented in polygon meshes, triangle meshes, point clouds, and voxel grids. The polygon meshes also contain semantic information of objects. Based on RealCity3D, we explore city-level generation and perform two benchmarks including building footprint and building shape generation. Through these benchmarking experiments, we demonstrate that our dataset poses novel challenges to existing data-driven generation methods on a city-scale scene. In the near future, we will extend our dataset to include more cities and more shape categories, such as roads, bridges, etc. We hope the RealCity3D dataset can accelerate the research community's work in developing deep generative models for large-scale generation.

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
