# OpenReview forum: "RealCity3D: A Large-scale Georeferenced 3D Shape Dataset of Real-world Cities"
_NeurIPS.cc/2021/Track/Datasets_and_Benchmarks/Round1 — Submitted to NeurIPS 2021 Datasets and Benchmarks Track (Round 1)_

### Official Review · Reviewer_iTSE · 2021-07-02
**Dataset with good potential, but insufficiently motivated and tested**

**Rating:** 4
**Confidence:** 3

**Strengths:**

+ The presented dataset has a high degree of novelty and originality. I agree with the authors, up to my knowledge, large-scale datasets of 3D city shapes are rare.


**Weaknesses:**

- The dataset is not sufficiently well motivated in my opinion:
1) lines 14 and 21: some potential applications areas are mentioned, related to remote sensing, urban planning, game design and autonomous driving. But references or details are not provided, so it is difficult to see the link and the use of this particular dataset to these areas.
2) While the dataset contains complex 3D polygonal models, the tasks in which the dataset is tested are 2D layout generation and cuboid shape generation. This makes the application area and use more diffuse, the full complexity of the dataset is not used in any of the tasks, it is difficult to evaluate if the gathered data is sufficiently large or diverse for its final goal.

- While the baselines and metrics chosen are reasonable, they are not related to the final goal of the dataset. The paper does not define clear baselines and metrics for the tasks of 3D city generation.

- The data is geo-referenced, but it is not explicitly written why this is relevant. None of the tasks designed use this information.

- Data curation is always time consuming for the authors and helpful for the rest. I am sure it is the case also here. But, it seems relatively straightforward compared to other datasets: the main steps are correcting aspects not meeting the standards and converting to different formats.

- The diversity of the data might be low, it just contains two cities with quite defined features. It is not motivated why just two cities is sufficient for the final goal of the dataset. Moreover, according to Figure 6, the majority of the data (over 95%?) comes from New York.

- In 3.3.3, it is said that this dataset suits "deep learning approaches such as voxel-based convolution neural networs and deep learning on unstructured point clouds", but no evidence or test is done in that direction.

- The tests done in Task 1 are confusing: reasonable baselines are chosen, a novel approach (AETree) is described with a very low level of detail, AETree outperforms all the baselines by quite some margin, and very little analysis is offered on that. If you developed such a nice novel method, shouldn't it be described with more detail?

- Again, in Task 1, the results in Figure 8 seem poor. In most baselines, buildings are overlapping. Only a few blocks are generated, very far from the final goal of city-scale generation. This results do not demonstrate how useful is your data.

- In 3.4, the paragraph only covers New York statistics.

**Additional Feedback:**

While I found the data collected and the targeted areas interesting and relevant, I find the dataset in a premature state to be accepted now. The specific use of the dataset in the application areas is not well defined, there are no references nor clear explanations. The data is tested on tasks that do not correspond to the final goal, and that makes the use of the data diffuse, there are not metrics defined for the final task. The results are confusing and far from the scale that should be expected from the available data.

I encourage the authors to work on these aspects, completing them would give insight on how this data can be used, how approaches should be benchmarked and what can be done with it that cannot be done with others.

**Clarity:**

+ The paper is written very clearly.
- In my opinion, there is a lack of details and insufficient analysis in the tests performed with the data.

**Correctness:**

+ The claims are correct in my opinion.
- As previously stated, while the tasks, baselines and metrics are reasonably chosen, they do not correspond to the final goal for which the dataset is designed.

**Documentation:**

+ The URL provided contains the dataset.
- The hosting, licensing and maintenance plan are quite vague, they state the intention of maintaining the data but no clear plans.

**Ethics:**

Not applicable for this work.

**Relation To Prior Work:**

- As the paper mentions the automotive industries, the authors should relate their work to synthetic datasets targetting autonomous driving. For example, the SYNTHIA dataset https://www.cv-foundation.org/openaccess/content_cvpr_2016/papers/Ros_The_SYNTHIA_Dataset_CVPR_2016_paper.pdf

**Summary And Contributions:**

The paper presents a dataset containing over 1M geo-referenced polygonal models of buildings from the cities of New York and Zurich. The dataset is curated from open data from the administration of both cities, and presented in several formats (polygonal and triangular meshes, point clouds and voxel grids). The data is tested on two tasks, city layout and building shape generation.

---

### Official Review · Reviewer_R8EN · 2021-07-04
**RealCity3D is an interesting city-level 3D building dataset but benchmarks are not good enough.**

**Rating:** 4
**Confidence:** 3
**Clarity:** This paper is well written and organi…

**Strengths:**

- This work propose a novel city-level dataset which collect the city 3D models in New York City and Zurich. It enlarges the city-level dataset community.
- RealCity3D have various data formats, e.g. meshes, point clouds and voxel grids, which are not provided by other city-level datasets. And the dataset provides 4 level of details of building objects.
- Two novel tasks, city layout generation and building shape generation, are proposed to explore the potential research on RealCity3D dataset. These two tasks may contribute to the community on urban planning.

**Weaknesses:**

- This work proposes a city-level dataset including 3D building shapes while I find that the building objects shown in Fig. 5 are synthetic 3D models. This paper compared RealCity3D dataset with SpaceNet and RoadNet that includes images from real scenes. Besides, home-level dataset like ScanNet also collects data from real scenes. I think it is better to explain the superiority of these synthetic building shapes over data collected from real scenes.
- Actually, urban planning is a complex task while the evaluation metrics are too simple to make a convincing evaluation.
    - For the city layout generation task, Overlapping Area Ratio is adopted. It only considers the overlapping area between prediction and ground truth while I believe the footprints of buildings can have totally different shapes (e.g. circle, triangle, rectangle, etc.). Besides, the footprints have their own orientations which are also not considered in this benchmark. The OAR can not measure such problems.
    - For the building shape generation task, I find this benchmark only provide metrics for point cloud generation but lacks of metrics for other three types of data formats. It is an obvious drawback since the advantage of this dataset is providing multi-types of 3D building shapes.
- The motivation of designing these two tasks is kind of weak. It should be explained in detail.


**Additional Feedback:**

The RealCity3D dataset is a novel city-level dataset which is an interesting attempt and may facilitate the urban planning research with deep learning technology. The benchmarks should be designed in a more appropriate way to demonstrate its potential in future research.

\* The comments are detailed, and this work can be improved after the discussion. I suggest that authors could submit the revised version to the 2nd round.

**Correctness:**

The dataset is well constructed. The evaluation methods and experiment design need to be improved in a more appropriate way.

**Documentation:**

The documentation of RealCity3D has sufficient details. It includes the link to download the whole dataset and has explicit instructions and files to help us access the dataset.

**Ethics:**

It seems that there is no obvious ethic concern in RealCity3D.

**Relation To Prior Work:**

This paper discussed relationships with previous Object-level, Room-level and City-level datasets. Especially for city-level dataset, it proposes a city-level 3D shape dataset formulated in several new structures which is different from satellite images used in SpaceNet and RoadNet. Besides, it benchmarks the building shape generation task, on which above two city-level datasets can not be applied.

**Summary And Contributions:**

- This paper proposes a city-level 3D shape dataset which includes several formats, e.g. polygon meshes, triangular meshes, point clouds and voxel grids.
- Two tasks, city layout generation and building shape generation, are proposed to explore the research directions on top of RealCity3D.
- A tree neural network is designed for generating spatial data hierarchically for two tasks.

---

### Official Review · Reviewer_UK8s · 2021-07-04
**The dataset is in high quality with clear motivation.**

**Rating:** 7
**Confidence:** 4
**Clarity:** The paper is well written.

**Strengths:**

+ Large-scale LoD2 level 3D shapes with high variance are provided. The shape details are quite convincing, and I think this dataset will benefit the research community.
+ The shapes are processed into point clouds, meshes, and voxel grids, which is available for multiple application and representation.
+ Two tasks are presented and the baseline methods are well evaluated.

**Weaknesses:**

The task of city layout generation seems like a simple 3D extension of 2D layout generation, and I don't see the necessity of such extension to 3D shapes.  More discussion should be supplemented to compare the proposed task to other existing 2D layout generation tasks.

**Additional Feedback:**

None.

**Correctness:**

The design of the dataset and benchmark evaluation is sound.  The significance of the proposed two novel tasks should be better explained.

**Documentation:**

The source of the data is clearly illustrated in the paper.  The initial documentation, URL to access, and license have been provided.  The code to reproduce the evaluation is available.

**Ethics:**

No ethical issues were found.

**Relation To Prior Work:**

The references are sufficient.

**Summary And Contributions:**

The paper prensets a city-level shape datasets, which consists of over 1 million meshes, point clouds, and voxel grids for various city buildings.  The 3D data are with a high variance in geometric complexity and spatial layout styles, such as residential/historical/commercial buildings and skyscrapers.  Two tasks are designed including city layout generation and building shape generation.

---

### Decision · Program_Chairs · 2021-07-26

**Decision:**

Reject

**Comment:**

There are two major concerns about the paper. First, as pointed out by the reviewers, the proposed benchmarks look incomplete compared to the novelty claim of the dataset. Even though generating full 3D city generation is hard to model and evaluate, we would expect the benchmark paper to shed light on the possible full solutions to show the quantitative difficulty. Second, the proposed AETree received negative reviews from ICLR 2021: https://openreview.net/forum?id=3NG1WgOn0y2. There was no final decision because the paper was withdrawn. However, the concerns raised by ICLR reviewers are not clearly addressed in this submission.